# The Adriatic Catholic Marian Pilgrimage in Nin near Zadar as a Maritime Pilgrimage

Mirela Hrovatin

Department of Sociology, Catholic University of Croatia, 10000 Zagreb, Croatia; mirela.hrovatin@gmail.com

**Abstract:** Following the general approach to pilgrimage as established by anthropologists and other scientists, the paper analyses the pilgrimage in Nin to Our Lady of Zečevo. More specifically, this pilgrimage will be observed as a maritime pilgrimage, following relevant recent research. Based on the oral story about the apparition of Virgin Mary to a widow, the statue of Mary is transported from Nin in a boat procession via sea to a mediaeval church on the nearby uninhabited island of Zečevo. Pilgrimage practices include many sensorial and symbolic practices, so it will be analysed from several points of view and more than one theoretical approach, including the relational approach and mobility turn, applied also to maritime pilgrimage with a reflection on influence of tourism on pilgrimage activities, especially in the Mediterranean. The paper relies on the field research from 2020–2023 in Nin near Zadar in Croatia which has been supported in part by the Croatian Science Foundation under the project 'PILGRIMAR' (UIP-2019-04-8226).

**Keywords:** Marian maritime pilgrimage; Adriatic; Mediterranean

## 1. Introduction

Each pilgrimage, observed as a part of culture in a broader analytical sense within the scope of cultural anthropology, is a special case (Eade and Sallnow 1991, pp. 2–3). It closely connects to the history and changes throughout time in a specific local community and many different approaches and interpretative frames can be applied to it, as it cannot be understood in its entirety (ibid.; Turner and Turner 1978, p. 148; Coleman 2002, p. 363). This paper approaches the pilgrimage to Our Lady of Zečevo in Nin near Zadar in Croatia as a heterogenous phenomenon (Eade and Sallnow 1991, p. 2; Coleman 2002, p. 357), incorporating more "discourses with their multiple meanings and understandings" created by pilgrims, local inhabitants, religious representatives, tourists, visitors and so on (Coleman and Eade 2004, p. 5; Coleman and Elsner 2003). In this sense, the pilgrimage in Nin will be analysed from several points of view and more than one theoretical approach. Throughout the text, an ethnographic account with a thick description of selected information (Geertz 1973, p. 27) with historical specificities (Asad 2002, p. 116) will be given in order to meet analytical questions and debate on their different meanings. This pilgrimage will be observed and interpreted from the aspects of person (body), place, text (Eade and Sallnow 1991, p. 3) and movement (Coleman and Eade 2004). This aspect will be broadened by the relational approach (Ingold 2000; Sidorkin 2002) and mobility turn (Urry 2002) connected to nature and reflected in "processes, effects and therapeutic mechanisms" of pilgrimage including embodiment, nature walking and social significance (Jørgensen et al. 2020, pp. 34, 44). In order to explain some of the workings and meanings of this pilgrimage beyond the new paradigms, older structural and phenomenological approaches will be used, especially the notions of communitas and liminality (Turner and Turner 1978), combined with the notion of how societies remember through traditional enactment connected to collective memory (Connerton 1989). The paper will also address "multivalent connections among religious, political and economic processes" (Coleman and Eade 2018) present in this pilgrimage from its first mention in 1516 until today. This will include reflection on UNESCO intangible

cultural heritage safeguarding policies and diaspora issues (cf. Coleman and Eade 2018, p. 4), as well as relations between religious festivities and tourism (Boissevain et al. 1979; Boissevain 1996) in this pilgrimage within the Mediterranean context (Albera 2006).

The pilgrimage to the small island of Zečevo will be analysed more specifically as a maritime pilgrimage which includes using sea vessels (boats) to cross the path through the sea with the sacred object to the location with historical or folklore connection with the sacred object or place (Katić and McDonald 2020, p. 3). The wooden statue of Virgin Mary is transported by a boat from the parish church of St Anselm in Nin on the 5th of May and 5th of August followed by other boats steered by the local people. The path from the church includes the passing of the procession with the statue through the streets of the town towards the little harbor, the travel across the sea to the small island of Zečevo and back, and the procession around the small church with the statue during the Holy Mass on Zečevo. In addition to by boats, pilgrims come to the island on foot. The pilgrimage is based on the official story about the apparition of Mary to a young widowed shepherdess in 1516 (Jelić 1900).

The methods used have been open interviews with local inhabitants, Catholic Church representatives, representatives of the local tourist office, city authorities, as well as pilgrims coming both from the local community and outside of it, from places surrounding Nin and further, from Croatia and abroad, mostly diaspora. One of the methods used was participant observation in which I engaged as a Catholic and shared with interviewees some of my views on religion and pilgrimage, what in part helped gain more in-depth view of the workings of the whole pilgrimage as well as life in Nin in general. Being aware that every ethnographic gathering of information and the following cultural anthropological interpretation are partial and in part subjective, I provide in this paper just one of the possible insights into the maritime pilgrimage in Nin. The Shrine of Our Lady of Zečevo is one of the places researched under the project "PILGRIMAR: Adriatic Maritime Pilgrimages in Local, National and Transnational Context" (UIP-2019-04-8226) from 2020–2024, supported in part by the Croatian Science Foundation.

## 2. Historical Overview and Changes of Pilgrimage in Nin

The statue of Virgin Mary in Nin is estimated to be made around the 14th–15th century and typical for the time as Mary carries the Christ Child (Strika 2011, p. 19). The small chapel within the church of St Anselm in Nin in which the statue is situated was built by the bishop Juraj Divinić after the apparition in the 16th century (ibid.). The statue was prior to that kept in the small chapel of Saint Mary, dated in 1335, which was a part of the monastery complex of hermits living at the island of Zečevo from the 13th until the end of the 15th century (Katić 2023a). The oral story was noted and accepted by the newly appointed bishop Divinić in 1516 and the Catholic Church officially established the pilgrimage the next year, as noted by the historian and archaeologist Luka Jelić in 1900. According to the story, a white-robed hermit first appeared to Jelena in Jasenovo near Vrsi on 9 April 1516 and ordered her to go to the church on Zečevo where she will be further instructed of what to do in Mary's honor so that God does not send his rage (whips) on Christians. On 4 May 1516, Mary herself appeared to Jelena in the church on the island of Zečevo with a message of what people should do. This was accompanied by the tears shed by the image of Mary, so that everybody present in the chapel could see the miracle and believe Jelena. Our Lady told Jelena that the faithful should pray to Her on the island of Zečevo and in the nearby chapel of St James so as to be "converted" to the right path, as well as delivered from their sins and the Ottoman threat. She supposedly said that whoever orders the mass to be held at the chapel of St James, makes Holy Confession, fasts whole Monday and takes the Holy Communion, She will provide them with a complete forgiveness of their sins each Monday in the chapel on the island of Zečevo.

The significance of a Marian pilgrimage in Nin should be observed with the awareness that the town was the first official settlement of the Croatian people from the 7th century after the Great Migration of Peoples (Strika 2011). The Croatian kingdom with the center in

Nin was approved as independent by the Pope John VIII in the 9th century. The kingdom encompassed the areas beyond the sea, towards the inland, the area from which pilgrims would later come to venerate Mary in Nin. Recent research points to one of the earliest representations of Mary in Croatia from the same period on a bursa-reliquary in Nin (Vedriš and Maraković 2021). Mary has been seen as a protectrice of the Croatian people, what has been continually confirmed by the Catholic Church in Croatia (cf. Hrovatin 2023) and expressed overtly by the Croatian people as interviewers say. The continuity of veneration from 1516 or earlier is unlikely due to many wars and political and social changes that took place in the following centuries (Katić 2023a). Certain is the confirmation that the veneration of Our Lady of Zečevo was established in 1516 and that pilgrimage as an organized festivity took place in Nin at the end of the 19th century as noted by Jelić (1900) and confirmed by older inhabitants (this research).

The sense of Nin as an ancient Croatian town of kings still remains in the imagery of both local people and people coming there from all over Croatia and diaspora from abroad. The apparition happened at the time of hardships the local people endured which were caused by the Ottoman and Venetian wars, and it served to strengthen Christian and Slavic identity, according to the earliest description (Jelić 1900). The important place of this shrine among other historical Marian shrines of the Croatian people is seen in that its catchment area in the past was more than 50 km inland, with the communities interdependent on the economy of sea and land. It attracted more pilgrims than any other shrine in this area, and certainly Mary as the strongest saint had her central place there compared to other saints venerated in surrounding places. Visiting each other during Church holidays and saint days was a usual practice and it was an important aspect of social life, maintaining relations among various local communities (cf. Christian 1989). One version of the oral story says that on 21 April, prior to the event on the Zečevo island, Mary appeared to Jelena near today's church of Our Lady of Jasenovo on a vine in the vineyard in Vrsi, which was at the time populated by the people who had cattle and sheep. Vrsi has until today been considered a shepherd and farmer's village in comparison to the town of Nin as a more urban place, connected to the old Croatian kingdom. To this day the hostility between the two neighbouring places remains, but they both participate in the pilgrimage, not unusual for many such cases in Europe.

According to the interviewees, until the beginning of the 20th century there was a transaction of goods among different communities within the catchment area, even during the pilgrimage. The fact that most of the sea travel on the Adriatic took place along the eastern Adriatic coastline due to the possibility of finding a safe shelter during bad weather, made this area beneath the Velebit mountain ideal for trade (Tomičić 2020). There were several trade fairs held in Nin during the year and a mill in Nin was in function where people from various places would come to exchange or sell their goods and mill their crops into flour. The same people would also come on pilgrimage in May and stay in the houses of the local people in Nin for three days. Mostly they would come for religious purposes, mainly absolution of sins and healing in the shrine, but also the socializing aspect was strong, including dancing their folk dances after the procession. Until well into the 20th century, the pilgrimage gathered all those people from the inland and the continental area, as well as fishermen and sailors' places and islands around Nin. Most interviewees talk about warmly receiving those people into their homes, and later, during pilgrimage and celebration, feeling a belonging to the same ethnic and religious community, including welcoming Orthodox Christians and other ethnicities.

One version of the oral story mentions the statue of Mary coming on a raft through sea to Nin after the Ottoman attack of the monastery on Zečevo and church bells ringing without a bell ringer to welcome her arrival. This story influenced its maritime aspect. However, there are no confirmations that the pilgrimage was at all maritime until the end of the 19th century (Katić 2023a). When comparing the description of pilgrimage in Nin from the end of the 19th century (Jelić 1900) to its versions from most of the 20th century which local people remember, several changes can be traced. One is the shift in the direction

of the land procession from the harbour near the mouth of the river Jaruga outside the town historical walls towards the harbour Gospin Mul/Muja (Our Lady's Harbor/Wharf) closer to the town. This change happened because of the development of Nin, building of new roads and change in trade in which the river no longer served as a safe harbor for those coming to the mill and to trade goods. The symbolical placing of the statue on the three ancient leftovers of the Roman pillars, more precisely three capitals, was replaced by the placement of the statue on only one capital near the main church before going to the new harbour. In the description of pilgrimage by Jelić (1900) larger sailing and trade ships came from the other nearby significant trading ports of Privlaka, Pag and Vinjerac testifying about a more pronounced maritime aspect of this pilgrimage at the time. The change from sailing towards motor ships and boats as well as tourism taking over as one of the main economies today on the Adriatic Sea (cf. Boissevain 1996), has led to a decrease in the number of sea vessels participating in the pilgrimage. In recent decades, only several smaller boats owned by local families accompany the statue across the sea, with a larger one carrying the statue.

Another change in recent decades is organizing the second, repeated celebration with procession in August, as people say "for the diaspora", what happened after Croatia gained its independence as a state in the 1990s. A larger number of boats participate in it with expatriates who come during summer to visit their families and spend their vacation in Nin, so they have the opportunity to participate in the pilgrimage. The pilgrimage was before the end of the 20th century organized on the 15th of August, Feast of the Assumption, but since many needed to get back home to USA, Australia and various countries in Europe, the locals decided to organize it sooner, on the 5th, when Mary of Snows is celebrated as well as the national holiday of Victory and Homeland Thanksgiving Day, which commemorates, among other things, the end of the 1990s war in Croatia.

While in St Mac Dara pilgrimage priests are overloaded with regular holy masses in the parish church and on the island (Katić and McDonald 2020, p. 9), masses and processions in Nin are organized to suit the established feast of Our Lady of Zečevo and its celebration. From the early morning there are several masses, the first at 6:00 a.m. for the blessing of pilgrims before the trip, and the later ones welcome pilgrims from afar who come during the day and locals who go by boat or by car (usually elderly and ill people go by car). During the Holy Mass on the island at 11 a.m., the priest blesses the boats and all other travel means, as well as people and nature. The mass is served by the parish priest with his co-adjutors guests from surrounding parishes whom he invites each year, as it was done in the past. The official feast of Our Lady of Zečevo is held on Monday preceding the feast of Ascension of Jesus, so the archbishop of Zadar comes to lead the Holy Mass in Nin which allows for the participation of a larger number of people than on the island. For some it is not exactly pilgrimage, but also commemorating the event of the apparition, so they say "We are going to the Apparition (Prikazanje)" when they talk about going to the island of Zečevo and also to the festivity on Monday.

Gender roles in traditional pilgrimage activities have also changed up to today, so the male sacristans clean the church in Zečevo and prepare a special flower wreath for the final festivity, which was the exclusive role of young unmarried girls several decades ago. They also organize some of the other activities, including preparing (dressing) the statue of Mary with the help of nuns, in addition to their regular role of coordinating the processions and transport of the statue by boats or by car. Although the priest is in charge, most of the activities and instructions are completed in agreement with the sacristans. The organizational problems in recent years arose from the overload with the tasks, and lack of transmission of the tasks to younger members of their families or congregation. In 2022, they forgot to place the thread with flowers onto the capital stone near the church, and as I was standing close by, waiting for the procession to make a circle and take some photographs, they placed it with my help as it was cumbersome to carry by one person. Yet, participating in the pilgrimage, walking on foot and in processions, singing and coming to the Holy Mass is still observed by the younger generations, mostly because the whole

families gather and go together. Additionally, a special prayer group consisting mostly of older married women is very active in maintaining different prayers before and after the masses, including rosary, litanies, Anima Christi and other. The church choir is very active too, mainly composed of married women who are not from Nin. There is also a Girls' Society of Our Lady of Zečevo, the task of which is to carry the banner with the image of Virgin Mary during the procession. When the girls from the Society are prevented to carry the banner due to everyday obligations, it can be carried by any woman, which was not so in the past when it was the exclusive role of unmarried girls.

When talking about going by boats to the island of Zečevo, local people describe it as something that is logical for them living near the sea. Unlike those at Dergh, locals who are fishing for personal needs in Nin do not identify themselves as fishermen, mostly because they are not registered officially as fishermen by the state authorities. Nonetheless, most interviewees talk about fishing smaller amounts and using fish as a part of their traditional, everyday diet, so that it goes without saying that they use boats also for fishing. They sometimes compare it to hunting game, among which they mention ducks and hares. According to the interviewees, hares are present also on the island of Zečevo reflecting its name—the island of Hare(s). Although in Galway there are some signs of motivation for continuation of the wooden boatbuilding as part of the need to go fishing (Katić and McDonald 2020, p. 7), in Nin it is not the case, as tourism has been taking over the local economy and overwhelmed the locals, so any additional strain, including the safeguarding of this know-how, would take time they cannot provide for now. The lack of interest has also hindered the inscription of the making of wooden boats into the national Register of Cultural Goods as intangible cultural heritage, and this further reduced the prospects of its continuation. Even with the inscription of the holiday of Our Lady of Zečevo in the Register 2013 as intangible cultural heritage, there has not been a significant change in the locals' attitude towards these connected traditions, so no new activities have been started nor some abandoned cultural practices revitalized, as proscribed in the document of the inscription (Resolution on pronouncing, 2013). So, it continues as a religious event with the similar pace as before the inscription. This is also due to the fact that the Tourist Board Nin suggested the inscription, recognizing it as a living heritage worthy of such a confirmation on national level, however without the active grassroots involvement of the local people and Church (ibid.).

However, the heritagization of this pilgrimage happens on other levels and the Tourist Board Nin leads the way. The Board invites cultural associations to come in their national costumes to pilgrimage and dance after the Holy Mass. It also invites media to cover the pilgrimage and announces the event as part of tourist attractions on their official site (The Church Holiday, s.a.). Interestingly, it has not been received by tourists massively, which is according to intangible heritage safeguarding policies a positive thing (Information Sheet, s.a.). Rather, tourists, while on their holiday, just stumble upon the procession through the town streets and thrillingly take photographs, some just observe the event, and some join the procession by walking, but only in Nin, not on Zečevo.

Going back to the significance of boat travel, the island of Zečevo is an important spot in the seascape as, according to the local people, it provides shelter to sailors and fishermen from bad weather, storms and strong winds (bora). Unfortunately, with a lack of archival sources, it remains only a guess that the wandering 13th-century eremite monks found a shelter there during their voyage and a secluded place to dwell and pray. The eremites built a church and monastery on the island of Zečevo and the oldest mention of Mary is connected to their presence (Katić 2023a). The church in St Mac Dara was also probably built by the hermit monk as described by Katić and McDonald (2020, p. 9). Unlike in St Mac Dara, if it does not take place by boats due to bad weather, the statue is transported from Nin in a van to the island of Zečevo and people come mainly by cars to the crossing from the land to the island and then walk several miles to the church there. The trip by boats takes around half an hour, a bit more than the one in St Mac Dara (Katić and McDonald

2020, p. 17), while walking on foot from St Anselm church in Nin to Zečevo takes around three hours with two main stops for prayer and rest.

Newer pilgrims' practices, some of which will be further explained here, do not stay confined only to Nin, but pilgrims, both those from Nin and those coming to Nin, regularly visit other shrines in Croatia and worldwide, depending on the occasions and their spiritual and social needs, but this goes beyond the scope of this paper.

## 3. Embodied Experience, Heritagization and Constructing Seascape within Pilgrimage in Nin

There are many similarities of pilgrims' motifs, feelings and processes in Nin with those so far described in literature (Badone 1990; Frey 1998; Jørgensen et al. 2020). After the arrival to the shrine in Nin, pilgrims embark upon a plethora of ritual practices, including walking around the statue of Mary displayed in the middle of the shrine three or more times, going underneath the carrier of the statue and giving gifts in money near the assigned place or close to the statue. They pray rosary, kneel, some go on their knees three times around the statue, or around the altar, they light candles, pay for the mass for their family and/or deceased ones and go to confession so they can participate in the Holy Mass later and receive the Holy Communion.

The pilgrims in Nin follow the path, which was established during centuries, based on the authorized oral story confirming it each year over again in space. The important two stops are the churches of St James and Our Lady of Jasenovo, which with the starting one in Nin of St Anselm and the final one of Our Lady of Zečevo on the island, make four churches altogether on the pilgrims' route. People rest near the two churches on the road, and some leave their rosaries on the vine near the church of Our Lady of Jasenovo where the apparition took place according to the story. Along the road, girls and women pick flowers in the fields, so called Our Lady's flowers (Latin: Ranunculaceae) near one well before the second stop. One pilgrim said about picking the flowers: "As much as it is tradition, so much it is the rest for the soul. Really, when you come to this field, you have a feeling that nothing is happening in the world" (Hodočašće Gospi od Zečeva, s.a., author's translation). Walking in this pilgrimage is also a way to connect with the space of the entire Nin area, as pilgrims going on foot pass near vineyards, meadows, fields and houses. One of the pilgrims usually takes their bicycle, but uses it only for return, while they walk towards the island pushing the bicycle along the way.

Along the road, a special type of friendship and fellowship is made as most pilgrims share with each other their religious and life experiences. In addition to this aspect, pilgrims in the interviews talk about rethinking their everyday life, and some their whole life, their hardships and sorrows, relationships with other people and so on. The prayers for a specific healing have significantly been reduced in recent years, and more prayers for the overall wellbeing of the whole family are prayed, as well as for the spiritual benefits, such as absolution of all (life) sins, building a better person according to the Biblical texts and Christ's teachings they hear about during the masses and in sermons. The older people pray also for the whole country of Croatia. In 2022, one lady said they pray for Ukraine because there is war, remembering the 1990s war in Croatia, and for world peace (Hodočašće Gospi od Zečeva, s.a.), the latter being one of the common recent instructions from the Church.

The older inhabitants also mention a different path that went until the middle of the 20th century near the river Jaruga and salt fields. At the time, the walking on land and sea travel were combined into one pilgrimage as described before. When pilgrimage took place at that first harbour near the river, as soon as the statue would be placed on the ship, pilgrims would wet their feet and legs in the sea and wash their face as part of traditional ritual practices with the meaning of the blessing and inviting health. No one washes in the sea with this belief today. I did not even wet my feet when boarding and disembarking from a boat, unlike what Katić and McDonald experienced (Katić and McDonald 2020, p. 17). However, in August, some go against the overall attitude that pilgrims should not bathe in the sea, mostly because they need to undress and show most of the body in a

swimming suit which is not acceptable to the Church. So, they take a swim even before the main mass on Zečevo, as well as afterwards. Some of the locals hold a grudge against the swimming pilgrims, although for rare tourists taking a swim they have no objections, as they do not expect tourists to know the acceptable conduct during the festivity. In May it is still too cold, so there are no swimmers then.

As on land, so on sea, boats pass along the way from Our Lady's harbour towards the church of Our Lady of Zečevo on the island which can be seen from the harbour. Joining the trip by boats is open for everyone provided there remains enough space after family members, relatives and friends have boarded. During the transition by boats to the island of Zečevo there is not much talking, unlike when walking, during which people talk a lot in between the prayers. The person navigating the boat is important to all onboard, as the passengers have to give their trust to him (it is always a male) in hope nothing will happen to the boat while at sea. However, it also depends on good behaviour of the passengers and not tilting the boat with sudden movements, of which we were warned by him before the journey last year.

The island of Zečevo is uninhabited, and this allows space for the mystification of this sacred place by pilgrims. It is always challenging for pilgrims to come there, and I also felt it on the way, especially when going on foot and crossing sea to get there. The clearance of the space, trees nonexistent due to strong bora wind, only smaller shrubs, pointy herbs and some wildflowers, with stone pebbles along the way. The silence because there is no traffic, no murmur of the people, except pilgrims, combined with some rushes of wind whistling and sea waves splashing. All this impacts greatly the mind and body, the senses and thoughts of the pilgrims, only to reveal the church at the end of the way. The church is the last post of the pilgrim's path and beyond it there is only sea. Similarly, coming by boats across the vast sea surface also touches pilgrim's senses. Splashing of the waves onto the sides of the boats sometimes sprinkles the pilgrim, rushes of wind blowing through hair and face while sitting in an open boat, and sun, if clouds are not hiding it, hitting directly on the top of the head. At the end of the sea way a small church reveals itself, beyond it only arid uninhabited land. Even coming by car is challenging, as the road is not friendly to the tires. In addition, crossing over an improvised wooden bridge sometimes results in a car falling into the shallow salty sea, not friendly to the metal body of the car. Into the shallow sea sometimes also step in pilgrims when the water rises during high tide. Those who know about this transitory path take their towels on the trip, take their shoes off once they reach the passing and purposely go barefoot through the water. I did it also, and it gives a sense of relief from walking and heat as well as transition to the uninhabited island at the same time. For me, it was the most liminal practice, more than the travel by boats on sea and passing by the fields on foot.

After the mass in the church on Zečevo, people socialize, and some have a traditional picnic. Interviewees describe how it was important to have lunch there after the whole day of fasting, walking and praying, as it would be too strenuous on body to go back on an empty stomach, especially on foot. Today it is not that everyone should leave as they came, so some combine going back by cars or by boats, although they arrived on foot, some who came by cars go back by boats, and other combinations.

## 4. Discussion

A significant influence on the attitudes of pilgrims have official discourses of a religion, and in case of Catholicism those can be Biblical texts, teachings, and sermons (Eade and Sallnow 1991, p. 3). The first major step towards establishing the pilgrimage in Nin was the recognition of the shepherdess and widow Jelena's vision as valid in 1516 by the Catholic Church (Jelić 1900). Those were the years after the devastation of the area during war between Venice and Ottomans, with some other political problems (Novak and Maštrović 1969, p. 542). War, disease and famine caused displacement of people from Nin (ibid.), and other religions besides Catholic were available for people to choose, including Orthodox Christianity and Islam. Among that turmoil, the official acceptance of the apparition and

establishment of pilgrimage resulted in a firmer connection of the area (cf. Maunder 2001, p. 32; Katić 2023b, p. 9) and brought people back to Nin and to the Catholic Church.

With the officially authorized oral story, the Church significantly shaped most of the practices of pilgrims on the pilgrimage in Nin, while retaining its important religious goals, such as Confession and Holy Communion. The influence of not only written texts of sermons and holy masses, but also of the oral story in its several variations in Nin, have been an important constructive element of the "ontology of the sacred and its epistemology" in this pilgrimage (Coleman and Elsner 2003, p. 4). Through those texts the Church influenced the believers and pilgrims to strive to behave in a proper, ethical and moral way, thus shaping people's worldview for centuries. These attitudes and feelings are present among pilgrims in Nin even today, similar as in other European societies shaped by Christianity (Badone 1990, p. 22). The sacred ideal pilgrims seek (Morinis 1992, p. 2) by achieving spiritual and bodily cleanliness can be met in the shrine and during pilgrimage, what many pilgrims mention. It is important for them to prepare well, both physically and spiritually, so they fast and go to confession. These organized activities during pilgrimage, including masses, confession, processions and other, "provid[e] a basic framework of activity into which particular groups and pilgrims can insert themselves" (Coleman 2000, p. 162). Together with traditional ritual practices, all those activities make this pilgrimage "a form of movement that is also a type of embodied replication" (ibid.) of what is considered the authentic and historical pilgrimage by its participants (Boissevain 1996).

The pilgrimage and shrine in Nin function as many in the world, as described and interpreted by many scholars so far, among which those mentioned in this paper, including (Turner and Turner 1978; Boissevain 1996; Christian 1989; Badone 1990; Frey 1998; Coleman 2000; Jørgensen et al. 2020; Katić and McDonald 2020). The differentia specifica from most described pilgrimages is that the pilgrimage in Nin is undertaken by travelling on boats to the nearby island, similar as in St Mac Dara (Katić and McDonald 2020), with the statue of Mary onboard. Commemorating the apparition and the oral story by going to the pilgrimage to the island of Zečevo, while transporting the statue of Mary across the sea and back, is an aspect of this maritime pilgrimage belonging to the ways in which memory is maintained and how societies remember through practice (Connerton 1989). There are some aspects of this pilgrimage that might point to a long tradition of Christian Marian festivities in this central part of the eastern Adriatic appearing parallel to and influenced by similar ones in the Mediterranean. The oral story mentions arrival of Mary to Nin on a raft and this detail is present in several other stories along the Mediterranean coast in places having a similar tradition of maritime pilgrimages, not only connected to Mary but also to other saints, for example in Boulogne-sur-Mer in France (Lim 2020; Katić and McDonald 2020; p. 3, footnote 3). It was not uncommon that various sea vessels sailing through the Mediterranean on sails and with oars would be wrecked, either during a bad weather or battle. In shipwrecks the images or statues would often be washed ashore and in many cases given special divine powers by the people who found them (Remensnyder 2018, p. 314). Most of those sites also have natural salt marshes nearby, including the one in Boulogne-sur-Mer in Camargue (cf. The history of the Aigues-Mortes salt marsh, s.a.), and several in Venice (cf. Caorle, the small Venice, s.a., Maria, 2020) which opens other analytical questions which will not be addressed here. The idea of moving Mary represented by a statue on a boat each year to commemorate this miraculous event goes into the realm of sharing of Mary among the people in the Mediterranean and "sacralizing" her statues "through prayer and pilgrimage" (Slyomovics 2020). So, there still may be further research completed to reveal the pathways of veneration of Mary and other saints as boat-driven statues, especially along the Adriatic coast.

The maritimeness of this pilgrimage is to be found also in the relation towards the sea by the local inhabitants who venture on a trip to the shrine by their boats, which they usually use for fishing and transport to short distances around Nin, similarly as in St Mac Dara (Katić and McDonald 2020, p. 7). The changes in the way of life of the local people, mostly tourism taking over as the main economic income (Boissevain et al.

1979), reflect in the pilgrimage in Nin itself and its maritime aspect. The reduction in the number of boats participating in the sea procession and lack of willingness to travel by boats even at the slightest windy weather, all point to the change in the perception of the environment and oral story about Mary that fades among the younger believers since the 2000s. Katić and McDonald describe that many pilgrims come to St Mac Dara only out of tradition and curiosity, and are less motivated by their religious beliefs (Katić and McDonald 2020, pp. 19–20). The interviewees retold how fifty or more years ago a priest who was also a sailor in his youth encouraged the participation of boats in the pilgrimage in Nin. This reveals how the Church and its representatives can significantly shape these non-institutional practices, but also how even those aspects of pilgrimage which make it specific can be fragile and start to change during time. Heritagization might turn the whole issue around and motivate the younger again to participate in the event by boats, but this for now remains only a possibility.

Unlike the locals in Dergh (Katić and McDonald 2020), the people in Nin do not identify as fisherman. However, their activity of fishing small amounts of fish for family needs whenever they have the opportunity makes them people who live from and by the sea. In this sense, their participation in the procession by their boats across the sea fits into the symbolic aspect of their local, as well as of broader Mediterranean culture (Albera 2006; Boissevain et al. 1979). When asked about the posts where they usually fish, the locals named a few, among which also those around Zečevo. Zečevo is not that important for fishing, nor is it the border between some island communities. Rather, it is an important passage through the eastern Adriatic Sea from both the continent and from south (Kozličić 2000), also providing a shelter in case of bad weather (Katić and Blaće 2023). It served also for navigational purposes as many other sacred buildings (temples, churches and so on) on the Mediterranean Sea (Gambin 2014, p. 9). There is the possibility for all interested to be transported by the locals for free by boats as part of the sea procession, which is more pronounced in St Mac Dara to which there is no land path (Katić and McDonald 2020, pp. 18–19). The combination of trust, excitement and going almost to the unknown in a small boat is felt by those who embark on the sea trip for the first time, what Katić and McDonald also experienced at St Mac Dara (Katić and McDonald 2020, p. 15). In this sense a type of communitas is formed (ibid., Turner and Turner 1978), with the shared feeling of presence in a fleeting moment in time while moving towards the sacred or special place on the island (cf. Coleman 2002, p. 358) either by boats or on foot. Other boats joining in the pilgrimage are those of expatriates, yet most of them just arrive on the island of Zečevo without participating in the boat procession (cf. Boissevain 1996, pp. 6–7), as do also some of the tourists passing by the island.

Although it appears a structured pilgrimage, managing the pilgrimage in Nin depends on only several parishioners, the priest and nuns who engage in various preparation activities. The problems of passing on the activities to the younger generations are not clearly seen, but sometimes hinder some of the practices, for example organizing the boat procession. There is a difference between believers and non-believers visible in organization, the latter sometimes being only observers (cf. Coleman 2000, p. 159). Mainly local families closer to the Church in general organize and head the procession, carry the statue, carry the banners, use boats and manage other parts of the pilgrimage (cf. Boissevain 1996). Change in power relations, including gender roles (Maddrell and Scriven 2016 in Katić and McDonald 2020, p. 11; Dubisch 1990) among the members of Nin community during recent decades reflects in the preparation of flower decorations for the pilgrimage by male sacristans rather than young unmarried girls, carrying of Our Lady's banner by young women rather than unmarried girls, and older people organizing parts of pilgrimage with almost no transmission of the tasks to the younger members of their family or to the other interested parishioners. The interviewees mentioned there exists also a latent difference between locals and diaspora/expatriates who come once a year and participate in the celebration while also visiting their families (cf. Katić and McDonald 2020, p. 8), which

is why the celebration on the 5th of May is more for the local people (cf. Boissevain 1996, p. 13).

The uninhabited small island with the mediaeval church is also a differentia specifica of the two maritime pilgrimages, in Nin and St Mac Dara (Katić and McDonald 2020, p. 13). One part of the oral story describes how the monk in a white robe appeared to Jelena, obviously connecting the veneration to monks (ibid.). Katić and McDonald noticed that hermits living on remote islands are important for shaping maritime pilgrimages (Katić and McDonald 2020, p. 9). A hermit dwelling in mediaeval times is significant to people in Nin from historical point of view as probably the place of the origin of veneration of Mary (Katić 2023a), as well as a shelter for boats during bad weather and a passage in the sea along usual sea pathways. However, it is exactly what hermits mostly sought, this emptiness of the small island, that enables pilgrims even to this day to engage with more focus on their individual processes while journeying (Frey 1998; Jørgensen et al. 2020). The uninhabitedness of this island allows also for the two opposing communities in Nin and Vrsi to meet at a symbolically neutral terrain as well as for the meeting of numerous pilgrims coming from afar, from the surroundings of Nin as well as other parts of Croatia and world, mostly expatriates. In most cases, the feeling of coming home when coming to the shrine on Zečevo from any place in the world, including the nearby Nin, holds people together and gives them a sense of belonging (Coleman 2014, p. 288), while revitalizing the pilgrimage (Boissevain 1996, p. 12). The pilgrimage in Nin is an important opportunity to gather Croatians from diaspora in recent decades, especially after the Croatian state independence since the 1990s. Unlike expatriate tourists, other types of tourists are mainly observers who see this pilgrimage as another cultural and social event in the tourist destination, and do not collide with the locals in pilgrimage activities or "destroy" the event (cf. Boissevain 1996, pp. 6–7).

Rather than only "evoking time and place that no longer exist" to make the "right experience of the pilgrimage" (Katić and McDonald 2020, p. 13), in the opinion of the interviewees the participation in pilgrimage in Nin and on Zečevo extends the presence of the sacred there (cf. Feldman 2017). This allows for respect and worship through the memory of the apparition and commemorating the apparition event itself, including transport of the statue by boats (cf. Connerton 1989). In this sense the spiritual and bodily experience of the believers is authenticated and much is expected as they search for predictable experience (ibid., Coleman and Elsner 2003, p. 6; Morinis 1992, p. 21 in Katić and McDonald 2020, p. 13). For pilgrims, the repetition of the feelings and sensations from the last time they went on the pilgrimage (Jørgensen et al. 2020, p. 46) creating a certain sense of home through movement and memory (Coleman 2002, p. 364) is affecting their view of the significance of this pilgrimage. Having a picnic at the island of Zečevo, similar as the one in St Mac Dara (Katić and McDonald 2020, p. 19), additionally strengthens social ties, and so does the freedom to mingle and join one another on the way back by changing means of transport. Both occasions create communitas-like feeling among the pilgrims, as well as bring them closer together, similarly to what was observed in Saint Olaf pilgrimage (Jørgensen et al. 2020, p. 39).

The official participation of the priests and the Zadar bishop has been in service of the pilgrimage itself, and each year it authorizes over again both the oral story and the pilgrimage in Nin itself (Coleman and Elsner 2003, p. 6). With this official involvement in organizing confession, holy masses, processions and sermons with the newest messages from Vatican, the Church fully participates in the maintenance and continuation of the pilgrimage in Nin. Touching the statue while circumambulating it, taking flowers home for extending blessing from the proximity of the sacred, expectation of healing and so on, all make part of the pilgrimage, as many other studies have shown. Almost inevitable need or possibility to get wet once at sea, and the enticement of the water to humans, including religious purposes (Eade 2023), is part of the past maritimeness of the pilgrimage in Nin, while picking flowers and refilling water at the well is practiced even today. All those official and unofficial "trivia" (Coleman 2014, p. 289) say a lot about the following of the tradition,

while respecting their natural environment (cf. Jørgensen et al. 2020), learning about the practices believers consider proper, about prayers from own members of the family as well as feeling of full participation in (re)creating the event itself (cf. Connerton 1989; Boissevain 1996) and in paying respect to Mary. The awareness that Mary appeared there and the oral story that she arrived on the raft all contribute to the miraculous discourse and creation of the shrine as a pilgrimage place (Eade and Sallnow 1991, p. 17) with maritime aspects. Both travelling by boats and walking also connect pilgrims to the past (cf. Coleman 2000, p. 161; Katić and McDonald 2020, p. 23), as they go along the path of pilgrims before them and visit important churches connected to the oral story.

Fellowship on the road and on sea, various motives for embarking on pilgrimage, rethinking of one's own place in the world and relationship towards their surroundings (nature, city, village, job) and towards people (family members, partners, work colleagues) in many aspects correspond to the observations so far in recent pilgrimages around the world (cf. Jørgensen et al. 2020, p. 45). The "embodied experience-movement that effects sensorial, emotional and affective stimulus" has a significant effect on pilgrims and those who join walking with pilgrims (Maddrell and Scriven 2016 in Katić and McDonald 2020, pp. 11–12). In addition to that, the sense of faith and need to participate in the long tradition of this pilgrimage is still very strong among pilgrims in Nin. By choosing what to (re)tell and give one's own interpretation of the experience during pilgrimage (Frey 1998; Jørgensen et al. 2020), people in Nin shape themselves as pilgrims (cf. Coleman and Elsner 2003, p. 5). The type of presence within the nature while walking, including picking flowers, has already been noted in previous research as part of liminality and freeing from everyday structure (Turner and Turner 1978), with tendency to transform pilgrims or walkers into a more focused ones with better connection between body, mind and environment, including nature and social surroundings (Jørgensen et al. 2020). Similar to the practices at Saint Olaf Way (Jørgensen et al. 2020, p. 40), pilgrims in Nin share among themselves their personal stories, their experiences of the sacred not only connected to the pilgrimage, Mary and the shrine, but also to their everyday life in which the sacred is also present for them or during visits to some other sacred places or while venerating other sacred persons.

The "geographic catchment area of the pilgrimage" (Katić and McDonald 2020, p. 5) in Nin extended in the past to the inland due to economic and ethnic reasons, while today it has shrunk to the surrounding places, not further than 20 km in range. However, the pilgrimage attracts diaspora and tourists, so the catchment area symbolically extends to the whole world. In this sense, this pilgrimage shows how changes in society locally and changes in the way of life globally reflect within it as in other areas and aspects of the local life and place (Eade and Sallnow 1991). Additionally, the connectedness of historical ways of trade of goods is reflected in the catchment area of the shrine and pilgrimage. The displays of national symbols, ethnic belonging, belonging to Christianity and so on, show how political economy can be important in shaping local shrines and pilgrimages (Coleman and Eade 2018).

Heritagization within the pilgrimage in Nin is visible in the waning of the overt display of identity via symbols of the national costumes, banners and on boats, while at the same time participating in the pilgrimage because it is a religious tradition, as interviewees say. This would correspond to the observance by Katić and McDonald that "folk beliefs being performed at the pilgrimage have diminished in importance over time" (Katić and McDonald 2020, p. 6). However, unlike the pilgrims in St Mac Dara, the faith in the sacred still governs a majority of the pilgrims coming to Nin, so the remnants of the old practices are being accompanied by "festive, social, identitarian, and economic aspects" rather than "overcome" by those (Katić and McDonald 2020, p. 7). In addition, the overt showing of faith through participation in the official and unofficial religious and social practices, inward prayers and so on, is still very much present in the pilgrimage in Nin, contrary to what Katić and McDonald pointed out for the St Mac Dara pilgrimage in which mostly socializing and leisure activities take over (Katić and McDonald 2020, p. 10) and Perast where religious aspect is vague (Katić 2023b, p. 10).

Regular believers, non-believers, priests, nuns, the prayer group, Girls' Society of Our Lady of Zečevo, the members of the choir, sacristans, tourists, expatriates and others, in different ways join in and confirm heterogeneity of the pilgrimage in Nin. So, multivocality and heterogeneity of the shrine, the image of Mary, and this maritime pilgrimage in Nin, testify how they can "reflect and affect aspects of people's lives" (Coleman 2002, p. 360), and that communitas, social cohesion, liminality and contestation can co-exist in a wider area connected to one pilgrimage (ibid., p. 361). It is also not always a Turnerian anti-structure of the society (ibid., p. 356) but rather a type of social, religious, and cultural (meta)structure existing for short period of time in a restricted space connecting people via repeated religious, bodily and metaphorical practices to the past (Turner and Turner 1978; Connerton 1989). At the same time, the pilgrimage can absorb in its structure continual changes through time, which is visible in Nin. The comparison of maritime pilgrimages (Katić and McDonald 2020; Katić 2023a) with the one In Nin, shows that even seemingly similar pilgrimages combine differently their historical, environmental, societal, economical, religious, and other aspects through time and space (Eade and Sallnow 1991, pp. 2–3).

## 5. Conclusions

The gathered data on the pilgrimage in Nin in Croatia, as well as its comparison with the maritime pilgrimage to St Mac Dara in Ireland (Katić and McDonald 2020), show how important coastal and island history and seascape are to the creation of this specific type of maritime pilgrimage. Maritime pilgrimages show some specificities regarding the connection to the sea and how it relates to the land, yet at the same time they comprise many aspects present in the pilgrimages already analysed in different parts of the world and world religions. They contain expressions of identity, embodied experience in connection to the sacred, nature and landscape, in this sense also seascape, socializing, practiced religion and overt showing of faith, inward journeys of individuals, openness of the pilgrimage to receive various types of pilgrims, including expatriates, as well as just travelers, tourists and passers-by, and so on.

The aspects of the importance of sea as a source of food, its vastness allowing transport of people and goods, including boats as means of travel, and its imagery as a part of the broader sacred creation, including its juxtaposition and connectedness to the similar importance of land, its imagery and safety, all become inscribed through centuries in the symbolism and practice of a maritime pilgrimage such as the one in Nin. The recent processes of heritagization and touristification of this maritime pilgrimage make it an even more complex phenomenon for the analysis and open various analytical and interpretative questions.

Semantics of the place is, among many other cultural aspects, built upon the spiritual and social significance of the shrine, with constant changes during time, yet the matrix of the pilgrimage tries to be maintained by the pilgrims and the Catholic Church. The matrix consists of transporting the statue of Mary to the island of Zečevo situated remotely from the inhabited part of Nin and Vrsi, as well as the official Church ceremonies, while unofficial practices of pilgrims change during time, including even the transport of the statue and pilgrims by boat. The tension between two communities, Nin as the one more oriented to the sea and Vrsi as the other one more oriented to the land, remains part of the imagery of this pilgrimage.

This research shows that pilgrimage as a part of living culture in one area remains a fertile field for further research, and maritime pilgrimages open some new questions and workings both within the local community connected to the pilgrimage site and natural surroundings as well as among pilgrims who come to visit the shrine.

**Funding:** This research was co-funded by the Croatian Science Foundation. Under the project "PILGRIMAR: Adriatic Maritime Pilgrimages in Local, National and Transnational Context" (UIP-2019-04-8226) from 2020–2024.

**Institutional Review Board Statement:** The research was conducted in accordance with the Ethical Code of the Croatian Ethnological Society (available at: https://hrvatskoetnoloskodrustvo.hr/o-nama/eticki-kodeks/ accessed on 13 February 2023).

**Informed Consent Statement:** Informed consent (both written and oral) was obtained from all subjects involved in the study.

**Data Availability Statement:** The data presented in this study are available on request from the corresponding author. The data are not publicly available due to agreement with the author stated in the informed consent of the interviewed persons.

**Conflicts of Interest:** The author declares no conflict of interest.

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
