# Peer review of "The Adriatic Catholic Marian Pilgrimage in Nin near Zadar as a Maritime Pilgrimage"

_religions, doi:10.3390/rel14050679_

Round 1

Reviewer 1 Report

This is a very interesting article. However, the authors do not report on the research methodology of the study, which makes the article seriously lacking in normativity.

Secondly, the references in the article are very old, perhaps historically, but why has there been no similar research in recent years?

Thirdly, the conclusions of the article do not seem to be well supported by the analysis section.

Fourthly, what is the contribution of this study?

Author Response

Dear Reviewer 1,

thank you for your suggestions. I have rewritten the whole article to make it more a case study, which might provide better evidence for my theoretical ideas. I remain open for further suggestions.

Best regards

Reviewer 2 Report

The article can become an interesting one after being much more worked by the authors.

The authors, in a large part of the article, limit themselves to findings, many of which are mere speculations, without ever properly justifying, at least in scientific/academic terms, how they arrive at many of the statements they make.

Author Response

Dear Reviewer 2,

thank you for your suggestions. I have rewritten the whole article to make it more a case study, which might provide better evidence for my theoretical ideas. I remain open for further suggestions.

Best regards

Reviewer 3 Report

This paper has considerable potential since it deals with an important research topic within pilgrimage studies and draws on original empirical research within an international research project. However, it needs to consider carefully some crucial theoretical issues and recent substantive research. It also needs considerable improvement in terms of the English text - for example, the meaning of a large number of sentences is unclear and definite and indefinite articles are missing. 

The key issues to consider are:

1. explain in greater detail how maritime pilgrimage contributes to pilgrimage studies in general.

2. explore more deeply debates concerning the Mediterranean as a cultural region. Since the 1990s there has been considerable criticism of this notion so there has to be a clearer justification for emphasising the idea of the Mediterranean as a cultural region. 

3. the case study shows the important role played by folklore so the contribution of folklore studies more broadly will provide a stronger theoretical base and raises an important question about the relationship between folklore studies and ethnographic focus within pilgrimage studies. 

4. the case study will gain wider interest if it looks beyond the Mediterranean to maritime pilgrimage in northern Europe, for example. Such a wider consideration would help us to see what is distinctive about this case study and other pilgrimages in the Mediterranean and what they share with maritime pilgrimages elsewhere in Europe. 

5. The text must be improved in terms of English.  

Overall, then, a promising paper which needs considerable improvement before it can be published as an academic journal article.  

Author Response

Dear Reviewer 3,

thank you for your suggestions. I have rewritten the whole article to make it more a case study, which might provide better evidence for my theoretical ideas. I remain open for further suggestions.

Best regards

Round 2

Reviewer 1 Report

Although the article has been extensively revised, I believe it is still not up to the level of publication and I am very sorry that I cannot support the publication of the article.

Author Response

Dear Reviewer 1,

I have started from other theoretical approaches, so practicaly the whole article is different now. 

Thank you for your patience.

Best regards

Reviewer 3 Report

The revised version is more focused and has considerable potential, while the research is original. However, its limitations become much clearer. As my comments on the text show, the paper claims to approach this particular pilgrimage from the perspective of cultural anthropology but what it actually does is to provide an historical description with a brief reference to recent developments. Anthropological analyses of pilgrimage have focussed on contemporary case studies, leaving the discussion of how pilgrimage was practised during the Middle Ages, for example, to professional historians. Recent analyses of pilgrimage across land and sea by social scientists have again focussed on their recent development drawing on ethnographic methods.

My advice would be to focus on the recent development of the Nin pilgrimage and link it to a stronger theoretical discussion of such key concepts as maritime pilgrimage, the Mediterrenean as a cultural region - see the important debate involving Jeremy Boissevain and others in Current Anthropology (1979) which has not been cited - Boissevain's work on ritual change in Malta during the 1970s and 1990s, a discussion of the similarities and differences between terrestrial and maritime pilgrimage which considers examples such as the Irish one referred to in the paper as well as the St Sulliva maritime pilgrimage in Norway and a clarification of such key concepts as rites of passage and communitas.

The paper needs to follow the structure of a journal article rather than a volume chapter so a separate Discussion section needs to be provided. Also the English needs considerable improvement. I have edited the abstract and the first paragraph to show you what is required. 

So, the paper has promise but needs to engage more deeply and thoroughly with empirical research on contemporary pilgrimage and relevant theoretical concepts and general debates. 

Author Response

Dear Reviewer 3,

I have started from other theoretical approaches, so practicaly the whole article is different now. 

Thank you for your patience.

Best regards
